# Influence of a Nano-Hydrophobic Admixture on Concrete Durability and Steel Corrosion

**DOI:** 10.3390/ma15196842

**Published:** 2022-10-01

**Authors:** Jingshun Cai, Qianping Ran, Qi Ma, Hao Zhang, Kai Liu, Yang Zhou, Song Mu

**Affiliations:** 1College of Materials Science and Engineering, Southeast University, Nanjing 211189, China; 2State Key Laboratory of High Performance Civil Engineering Materials, Jiangsu Research Institute of Building Science, Nanjing 211103, China; 3Jiangsu Subote New Materials Co., Ltd., Nanjing 211103, China

**Keywords:** concrete durability, reinforcing steel corrosion, chloride transportation, hydrophobic admixture, nano material

## Abstract

Steel corrosion is major reason of the deterioration of reinforced concrete structures. Decreasing the transportation of erosion ions in concrete is one of effective methods to protect the steel from corrosion. In the present work, a novel nano-hydrophobic admixture is introduced to improve the ion-diffusion properties and the corrosion resistance of reinforced steel. Compared with unmodified concrete, the nano-hydrophobic admixture effectively decreases the water adsorption, water evaporation, and chloride ions transport in a concrete structure, and then improved the concrete’s durability. The concrete’s water adsorption decreased more than 78%, and the initial corrosion time of reinforced steel is prolonged more than one time by treatment with the nano-hydrophobic admixture. The inhibition penetration of the medium in concrete modified by hydrophobic nanoparticles is the key to provide the protective properties of steel reinforcement from chloride erosion.

## 1. Introduction

Concrete is the most widely used construction building material. It is widely used in civil engineering, such as housing, agriculture, roads, bridges, water conservancy facilities, underground structures, nuclear power plants, and marine development. As the global climate changes and the field of construction expands, especially for the harsh service environment, the deterioration of concrete structures becomes more and more serious, which brings the biggest challenge for sustainable social development in the future [1,2,3]. The main degradation of reinforced concrete is the corrosion of the steel bars embedded in concrete. In addition, sulfate corrosion, freeze-thaw damage, and carbonation are also major causes of concrete’s lack of durability.

Usually, the steel in concrete is kept passivated in the high alkaline environment, which can protect the steel from corrosion by forming an oxide film. However, when the penetration of chloride ions reaches a certain concentration this will destroy the passive oxide film, or carbon dioxide diffusion into concrete will neutralize the alkalinity of the aqueous solution in the concrete pores leading to the instability of the passive oxide film [4]. The start time of steel corrosion subjected to aggressive environments is mainly controlled by the transport properties of concrete, i.e., carbon dioxide permeability, chloride ion diffusion, and water absorption, because of the high porosity of the hardened cement pastes. The water within concrete plays an important role in these transfer processes, which provides an essential carrier and diffusion medium for these degradation processes and the migration of aggressive ions, or is even involved directly in these processes [5]. In practice, steel corrosion would not occur if water is prevented from wetting the porous structure of the concrete [6,7,8,9]. So, the improvement of the watertightness and impermeability of concrete plays a key role in the durability and service life of a building and is beneficial to improve the protection of the reinforcement steel.

At present, the improvement of the permeability of concrete is mainly achieved by improving the compactness of cement-based materials and optimizing the pore structure by reducing the water/cement ratio, and the addition of auxiliary cement-based materials or nanoparticles [10,11,12,13]. However, there are still some problems with these common methods, such as increasing the shrinkage of concrete, affecting workability, and making dispersion difficult in highly aggressive environments. Currently, hydrophobic- or superhydrophobic-modified cement-based material has obtained widespread attention because of its excellent medium transmission resistance and great application potential [13,14,15,16,17,18,19,20,21,22,23,24,25]. The hydrophobic methods of concrete include surface hydrophobic and integral hydrophobic treatments. Surface impregnated with silane or applied coatings are two normal surface preparation methods, which can make concrete less susceptible to water saturation, but poor weather resistance and easy abrasion limit their use [13,14,15,16,17]. Integral hydrophobic treatment has the advantage of not being easily affected by surface erosion, so this technique has been widely studied in recent years. Fatty acid and silane emulsion are both typical internal hydrophobic substances; in addition, hydration-responsive nano-materials or smart polymers can also be considered as newly hydrophobic admixtures [18,19,20,21,22,23,24,25]. The integral hydrophobic treatment can reduce water absorption approximately 80%, decreasing the aggressive ions’ permeability and improving the corrosion resistance of cement-based materials. However, there is less research on the corrosion behavior of reinforced steel in concrete treated by integral hydrophobic agents, especially for nanomaterials with hydrophobic properties. F. Tittarelli studied the silane-based hydrophobic admixture on the corrosion of galvanized reinforcing steel in concrete and found that the hydrophobic concrete is able to protect galvanized steel reinforcement from corrosion, while the improper incorporation of hydrophobic substances will increase steel corrosion because of a higher oxygen diffusion rate [6,7,8,9]. Therefore, how to ensure that water, erosion ions, and gas transfer were inhibited without affecting steel corrosion is the key indicator to ensuring the service life of a reinforced concrete structure treated by integral hydrophobic agents.

For obtaining a more durable reinforced concrete structure, a nano-hydrophobic agent was incorporated into concrete. The effects of the nano-hydrophobic admixture on the corrosion behavior of reinforced steel and on concrete durability (water adsorption, chloride diffusion, and so on) were investigated. The nano-hydrophobic material is an aqueous emulsion, which can be easily dispersed in concrete. The purpose of this work is to explore the effect of a nano-hydrophobic admixture on the corrosion inhibition of steel in concrete, the resistance of water adsorption, and chloride ion transportation. The work will provide an experimental basis and application reference for the development of durable concrete.

## 2. Materials and Methods

### 2.1. Materials

The steel used in the present work is HRB 400 reinforced steel, the composition of which is listed in Table 1. Before testing, the surface of the steel was treated to become a plain round bar and then polished by emery paper (400, 600, 1000, and 2000 mesh).

Ordinary Portland cement PO 42.5 was provided by the Hailuo cement corporation. The chemical and mineralogical compositions of the cement are listed in Table 2. The nano-hydrophobic admixture used in this study is an aqueous anionic microemulsion of hydrophobic nano-silica (TIA, Sobute Co., Ltd., Nanjing, China). The average sizes of the nano-hydrophobic particles are ca. 60 nm. The effective content of the nano-hydrophobic is about 12%.

### 2.2. Mix Proportions and Specimen Preparation

The concrete with water-to-cement ratios (w/c) of 0.35 and nano-hydrophobic (TIA) admixture at dosages of 0, 10 kg/m^3^, 15 kg/m^3^, 20 kg/m^3^, 25 kg/m^3^, and 30 kg/m^3^ by the volume of concrete were prepared. The mix proportions of the concrete are listed in Table 3.

### 2.3. Specimens

The reinforced steel concrete specimens were cast into prismatic specimens with the side lengths of 40 mm after 2 min mixing. The fresh concrete was cast into cubic specimens with side lengths of 150 mm after 2 min mixing. All samples were vibrated on the vibration table for 1 min before being sealed and then were stored under an environmental condition with a temperature of ca. 20 °C and a relative humidity above 95% for 1 day. Finally, all specimens were de-molded and cured under an environment with a temperature of ca. 20 °C and a relative humidity above 95% to the designed ages.

For the reinforced steel corrosion test in concrete, prismatic specimens (40 × 40 × 100 mm in size) were produced (Figure 1). These prismatic specimens were reinforced with plain round rebar steel (φ 10 mm × 80 mm) embedded in one side of a specimen surface, the exposure surface depth from one side is 20 mm, and the rebar steel was connected with copper wire. To accelerate the corrosion of the steel bars, the wet–dry cycle test was carried out. The specimens were dried for 4 days and soaked in solution for 3 days in one cycle time. The drying temperature was 50 °C, and the soaking solution was 3.5% NaCl solution at 20 °C.

### 2.4. Test Methods

#### 2.4.1. Water Absorption Test

Water absorption tests were carried out using concrete specimens cured for 28 days, as according to BS 1881 part 122. For a representative sample, a set of three specimens was procured by obtaining a core 75 mm long when the thickness of the specimen was greater than 150 mm. The diameter of each core was 75 ± 3 mm. After drilling the core, the specimens were dried at 50 °C in an oven for 72 h; then each specimen was removed from the oven and cooled for 24 h in a dry airtight vessel. Each specimen was weighed and immediately completely immersed in the tank with its longitudinal axis horizontal and at a depth such that there was 25 mm of water covering it. The weight changes of the concrete were performed at 30 min, 60 min, 180 min, 300 min, 720 min, and 1440 min after the concrete began to contact the water. For each measurement, the specimen was quickly taken out, wiped with a dry towel to remove the free water from the surface, and then placed on the balance with the wet side up. The specimen should be put back immediately to continue the water-absorption test after weighing. The weighing process should be completed within 15 s. For each group of concrete, three specimens were tested and the average value was used as the experimental result.

#### 2.4.2. Evaporation Test

The specimens for the evaporation test are similar to those for the water adsorption test. The specimens were placed in an oven and refrigerator to keep the environment temperature at 50 °C, 25 °C, and 0 °C. The weight change with different times were recorded and calculated to determine the evaporation ratio.

#### 2.4.3. Chloride Diffusion Test

The chloride diffusion test was carried after dry–wet cycle experiment. The distribution of chloride ions from the depth of the concrete was detected by a chloride ion titration method, according to JGJ/T 322-2013. The samples, abrasive to powder, from different depths of concrete and immersions in acid solution to obtain the total chloride content. The chloride concentrations at different depths were recorded.

#### 2.4.4. Electrochemical Measurement

Electrochemical measurement was undertaken to validate the nano-hydrophobic admixture on the corrosion behavior of reinforced steel and the distinction of the admixture on the influence of concrete penetration or direct reaction with steel surface. Corrosion was detected in both the steel embedded in concrete and that immersed in the simulated concrete-pore solution.

The electrochemical corrosion behavior of the reinforced steel exposed to the chloride environment was evaluated by linear polarization measurements with respect to a saturated calomel electrode (SCE) as the reference and platinum as the counter-electrode. The electrochemical values reported in the graphs are averaged among the measurements carried out on three specimens of each concrete type during the immersion period.

The electrochemical test for steel in the simulated concrete pore solution was followed by EIS measurements. The simulated pore solution was composed of 3.5 wt.% NaCl, saturated Ca(OH)_2_, and a different concentration of TIA. The electrochemical parameter was detected after steel was immersed in the simulated solution for 24 h. This test was different from that of steel embedded in concrete because the steel electrode was directly in contact with chloride ions, and the kinetics of the steel corrosion process influenced by the nano-hydrophobic admixture was easily obtained. 

## 3. Results

### 3.1. Effect of Hydrophobic Admixture on the Water Adsorption of Concrete

The effects of immersion time and the content of nano-hydrophobic admixture on the water absorption of concrete were investigated; the results are shown in Figure 2. It can be seen that the water adsorption of concrete was effectively decreased by the incorporation of the nano-hydrophobic admixture. For a blank sample, the water absorption ratio with √t can be divided into two stages. Stage I is the rapid water absorption process, which approximately follows the √t law, but is not complete consistent with linear curves [5]. When the capillary pore was completely filled by water, the water absorption enters stage II, which is mainly gel-pore adsorption. During this stage, the water adsorption ratio is slower and gradually approaches saturation adsorption state.

When TIA was added into concrete, the water adsorption was effectively reduced as the concentration of the admixture was increased, and the water absorption ratio decreases even more. During the test time, the water adsorption amount and water sorptivity of concrete were both reduced by TIA; the saturated adsorption process was prolonged more than in specimens without TIA. The most water adsorption ration reduced by TIA during stage I was about 78%, and at end of stage II, the TIA also decreased the water adsorption ration about 63%. The results show that the TIA has the most effective influence on the capillary adsorption process.

For further investigation, the effect of TIA on the capillary adsorption process, different environment temperatures, and immersion time in the water adsorption of concrete by different dosages of TIA were explored (Figure 3). As the results show in Figure 3a, the biggest factor influencing water adsorption is the TIA concentration; the water absorption decreases more significantly when the dosage increases more, which is almost linear. The influence of temperature on the water absorption performance of concrete is insignificant, even when TIA is added.

With the prolongation of soaking time, the overall change trend of water absorption of concrete with TIA concentration does not change. The difference is that at 0 °C, the higher mixing amount of TIA will lead to more reduction in water absorption; the result shows that the water absorption ratios all decrease with the concentration of TIA increasing at different environment temperatures, while the influence of temperature on water adsorption was less.

Figure 4 is the water evaporation ration of concrete treated with and without TIA at different contents of TIA, immersion times, and environment temperatures. The results in Figure 4a–c show that temperature has a great influence on the rate of water evaporation at the initial drying stage. The lower the temperature is, the lower the rate of water evaporation. At high temperature, the water evaporation rate is faster. After long-time exposure, the effect of temperature on the water evaporation rate gradually decreases, such that after 5760 min of exposure (Figure 4d) the water evaporation rate at different temperatures gradually approaches.

The content of TIA has less effect on the evaporation of concrete during the initial exposure time (Figure 4a), while as the exposure time lasts, especially for long-time exposure, the inhibition effect of the hydrophobic substances on the evaporation rate is more significant, and the inhibition effect of the hydrophobic substances on the evaporation rate gradually decreases with the increase in dosage.

The influence of the exposure time on the water evaporation rate is also one of main factors in water evaporation. At the initial exposure time, the total evaporation amount is relatively less; as the time prolongs, the evaporation amount increases sharply for blank specimens. For concrete treated by a hydrophobic admixture, the evaporation ratio increases slowly, and more content of TIA also inhibits the water evaporation even more. After a longer time of exposure, the gap between the total amount of water evaporation becomes prominent, and the influence of the hydrophobic substance content on the water evaporation rate gradually increases. 

### 3.2. Chloride Ion Diffusion

Figure 5 shows the depth distribution curve of chloride ion migration and penetration in concrete after the dry–wet cycle. The results show that the distribution of chloride ions increases with depth within 5 mm of the surface of the specimen; then the total number of chloride ions reaches a certain peak value, after which the concentration gradually decreases. When TIA was added to the concrete, the distribution law of total chloride ions in concrete with depth was consistent with that in the benchmark specimen, but the total chloride ion content decreased significantly, especially within 5 mm of the concrete surface, where the total chloride ion content decreased by more than 50%. With the increasing of chloride penetration depth, the total chloride content decreases gradually. Results demonstrated that TIA can effectively retard chloride ion transformation and enrichment.

### 3.3. Electrochemical Measurements

Figure 6 shows the different dosages of TIA on the polarization resistance of steel bars in concrete under the condition of dry–wet circulation. The dry–wet cycle will accelerate the migration of water, chloride ions, and oxygen into the concrete, thus accelerating the corrosion rate of the reinforcement. After five cycles, the polarization resistance of the blank steel bar decreases significantly, and the polarization resistance is less than 100 KΩ, indicating that the passivation film on the surface of the steel bar has become blunt and the corrosion has begun to accelerate. With the incorporation of TIA, the initial corrosion time of the reinforcement is inhibited. As TIA content increases, the initial corrosion cycle of the reinforcement is prolonged. When TIA content reaches 30kg/m^3^, the reinforcement is kept in the state of passivation after 10 cycles. The initial corrosion time of reinforced steel is prolonged more than one time. 

The corrosion of the reinforcement in concrete is mainly affected by the concentration of chloride ions on the surface of the reinforcement, the concentration of humidity and oxygen in the pore environment, and various substances in the pore solution [1,4,6,7,8,9]. The clear conclusion is that TIA effectively inhibits the transport of water and chloride ions, which is the key to improve the corrosion resistance of the reinforcement. Whether TIA affects the electrochemical corrosion process of the steel reinforcement surface is unknown. To further analyze the influence of TIA on the corrosion behavior of steel bars and determine whether the TIA changes the electrochemical corrosion behavior of steel bars, the electrochemical corrosion behavior was studied in a simulated concrete pore solution (Figure 7). The results show that the electrochemical impedance of steel bars is slightly improved after TIA is added into the simulated concrete pore solution, which is different form the corrosion inhibitor [26,27,28]. The results demonstrate that the TIA has little effect on the steel surface; the steel corrosion resistance improvement was mainly because of the permeability resistance of enhanced concrete, the concrete pore structure, and the physical parameter change.

## 4. Discussion

As is well-known, water is the carrier of corrosive ions, and water adsorption is one form of driving ion transport. Normally, the water adsorption of concrete approximately follows the √t law [5,29]. There is little deviation in the present study; this deviation occurs mainly because of the swelling of C-S-H through interaction and the concrete microstructures continuously change during water sorptivity [5]. During this process, the water was mainly absorbed by the capillary pore; because of the stronger capillary negative pressure, the water absorption was quick. The incorporation of the TIA proportionally decreases the water adsorption, and such an effect increases quickly with the increase in the dosage of TIA [20,21,22,23,24]. Moreover, it is interesting to find that, different from the blank concretes, the TIA-modified concretes seem to follow the √t law, particularly at higher dosages of TIA. The results also confirm that the TIA can restrain the swelling of the C-S-H gel during the water adsorption process. These results are also evidence that the hydrophobic effect of the incorporated nano-hydrophobic admixture results in a significantly lowered absorption rate and amount because of the hydrophobic agent cover, or they fill the concrete pore then decrease the capillary’s negative pressure.

On the other hand, it is found that room temperature or a higher temperature has no effect on water absorption; this is mainly because the capillary pressure is the main driving force that impels water into concrete when the water was flowing as a liquid. As is well-known, the capillary pressure provided by a certain pore diameter r is shown in the follow formula [3]:(1)$$P=\frac{2\gamma \cos\theta }{r}$$
where *γ* is the surface tension, *θ* is the contact angle, and *r* is the capillary diameter.

At temperatures of 0~50 °C, the surface tension and contact angle of water change little in concrete; therefore, the temperature has less influence on the water absorption when the water is in a liquid and flowing condition. However, the pore distribution and pore size of concrete will change with the hydrophobic admixture [11,12,13], which causes water adsorption process variation with the concentration of TIA. 

However, the water evaporation process was significantly influenced by the temperature; even the TIA dosage is a main influencing factor. Water evaporation can enhance the water adsorption and result in more of the chloride content gathering, which is the why the dry–wet cycle is the most serious cause of damage to ion transport and the deterioration of concrete [3,18]. At the initial time, the concrete was saturated water, and TIA has little effect on the water loss. When a large portion of the water has been lost, capillary pressure will further inhibit water loss; that is the reason for the TIA suppression of the water evaporation after long-term exposure. Besides, the water evaporation process is a transition from liquid water to gaseous water; the temperature, evaporation area, and wind speed are principal influencing factors, with the regularly spaced distribution being much more effective in slowing the formation of the vapor tubes that trigger the evaporation process [29,30]. In the present test, the wind speed is identical; the differences are in the temperature and the structure of concrete pores modified by TIA. Changes of the concrete pore structure will influence the evaporation area and water-flowing process, which may be the cause of TIA inhibiting the water evaporation process.

The above results show that TIA can effectively inhibit the water flow into and out of concrete. Therefore, the chloride ion transmission was effectively suppressed. Chloride ion transportation includes three forms: capillary transport, infiltration, and natural diffusion [2,14]. Under the condition of dry–wet circulation, the capillary transport of chloride ions plays an important role [15]. Because TIA can significantly inhibit the hydrophobic concrete capillary water absorption rate and water loss rate, the rapid transmission and enrichment of chloride ions caused by water absorption and the water loss of concrete surface capillaries are effectively inhibited.

Because of water adsorption, water evaporation and chloride ion diffusion are restrained by TIA; therefore, the steel in concrete is effectively protected, but the steel surface is not changed by TIA. The inhibition of TIA on reinforced steel corrosion is mainly due to the improvement of the resistance of concrete medium permeability. For the future, the influence of TIA on the microstructures of concrete and the interface of steel and concrete should be study. Besides, the influence of this new hydrophobic agent on steel corrosion under other extreme service environments also should be paid attention to.

## 5. Conclusions

In this paper, the effects of a nano-hydrophobic admixture on concrete water adsorption, water evaporation, chloride ions transportation, and steel corrosion were investigated; the following conclusions can be drawn:(1)The nano-hydrophobic admixture exhibits a significant water adsorption inhibition behavior; the higher the content of TIA incorporated into concrete, the more water transportation-restraint performance was obtained. During the water adsorption process, the main influencing factor is the structure of the concrete pore modified by TIA, while the temperature has less effect on water transportation.(2)The nano-hydrophobic admixture also has an effective inhibition behavior on water evaporation, which is also influenced by environmental temperature. A higher content of TIA and a lower temperature favor reducing water evaporation.(3)Because the water adsorption and water evaporation process were both inhibited, the transportation of chloride ions was effectively restrained by the hydrophobic admixture. In the present investigation, the chloride ion concentration was reduced more than 50% compared with the blank concrete, which is normal marine engineering concrete.(4)The nano-hydrophobic admixture effectively prolongs steel corrosion by suppressing of chloride ion transport, which has less effect on electrochemical corrosion behavior when the chloride ions directly contact the steel surface.

## Figures and Tables

**Figure 1 materials-15-06842-f001:**
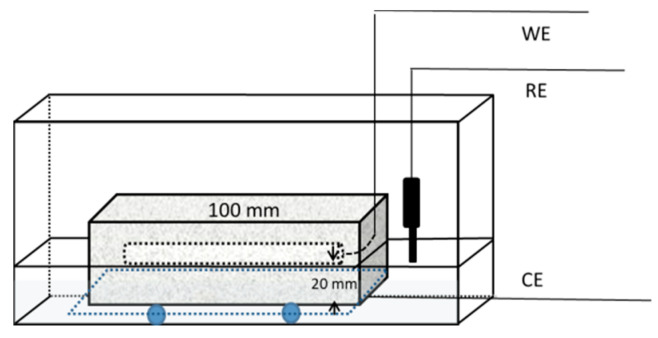
Schematic diagram of corrosion test.

**Figure 2 materials-15-06842-f002:**
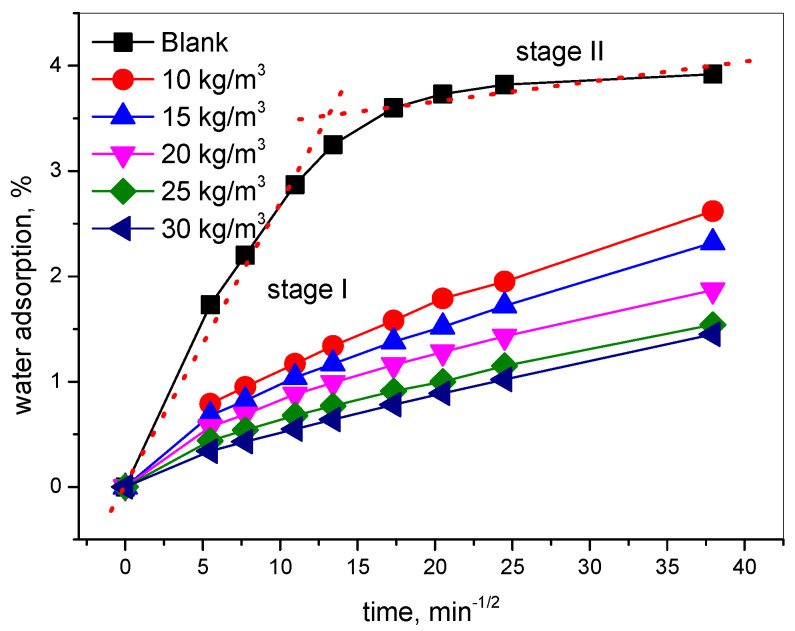
Water adsorption of concrete with different concentrations of a hydrophobic admixture as a function of test time.

**Figure 3 materials-15-06842-f003:**
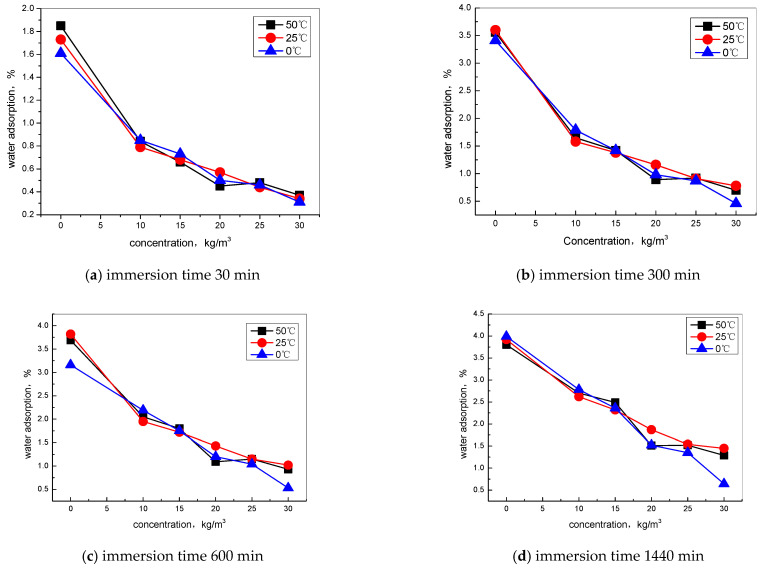
Water adsorption without and with different contents of TIA during different immersion times: (**a**) 30 min, (**b**) 300 min, (**c**) 600 min, and (**d**) 1440 min at environment temperatures of 0 °C, 25 °C, and 50 °C.

**Figure 4 materials-15-06842-f004:**
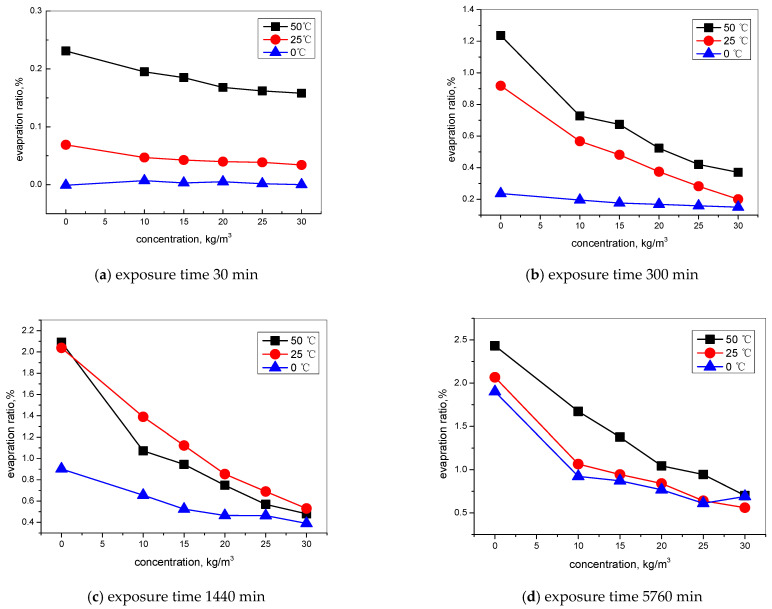
Water evaporation ration with different dosages of TIA and exposure times: (**a**) 30 min, (**b**) 300 min, (**c**) 1440 min, and (**d**) 5760 min.

**Figure 5 materials-15-06842-f005:**
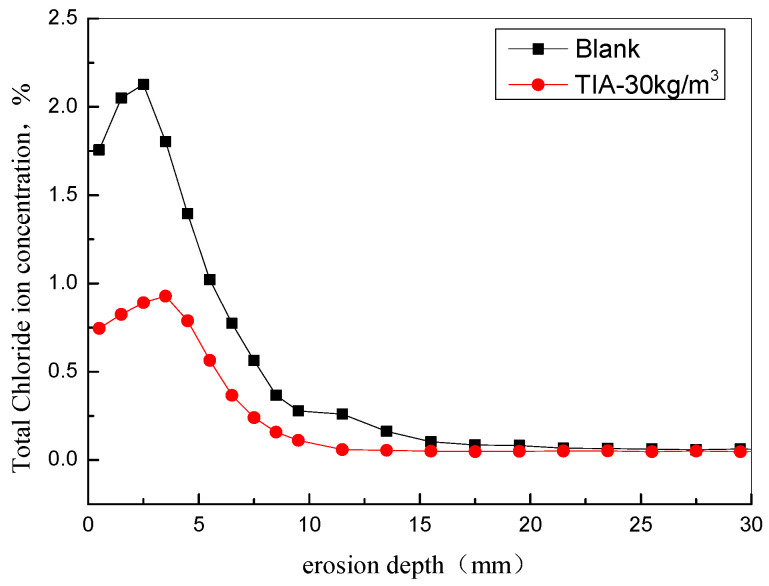
Total chloride ion depth from the concrete side with and without a TIA nano-hydrophobic admixture.

**Figure 6 materials-15-06842-f006:**
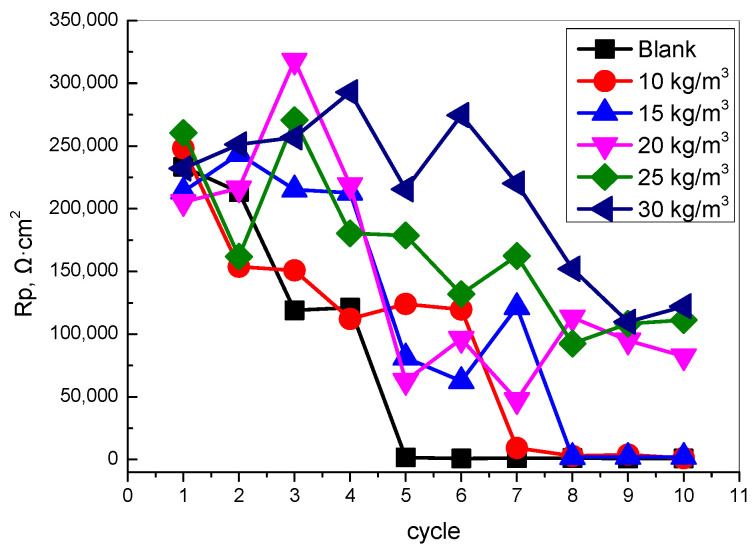
Corrosion resistance of reinforced steel in concrete with different nano-hydrophobic admixtures during the wet–dry cycle.

**Figure 7 materials-15-06842-f007:**
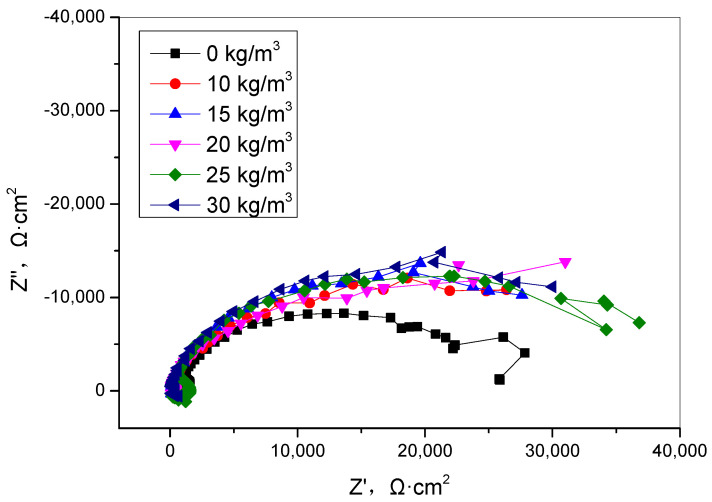
Effect of TIA on the corrosion behavior of steel in a simulated pore solution.

**Table 1 materials-15-06842-t001:** Composition of HRB400.

Composition	C	Si	Mn	P	S	Ceq
%	0.25	0.8	1.6	0.045	0.045	0.54

**Table 2 materials-15-06842-t002:** Composition of cement.

Composition	SiO_2_	Al_2_O_3_	CaO	Fe_2_O_3_	K_2_O	MgO	Na_2_O	TiO_2_
%	22.16	6.17	58.27	3.61	0.93	1.84	0.14	0.32

**Table 3 materials-15-06842-t003:** Concrete mixture proportions (kg/m^3^).

NO.	Cement	Flyash	Slag Powder	Sand	Gravel	Water	Water Reducer	TIA
**1**	220	110	100	730	1100	150	5	-
**2**	220	110	100	730	1100	110	5	10
**3**	220	110	100	730	1100	115	5	15
**4**	220	110	100	730	1100	130	5	20
**5**	220	110	100	730	1100	125	5	25
**6s**	220	110	100	730	1100	120	5	30

## Data Availability

Not applicable.

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
