# Peer review of "Influence of a Nano-Hydrophobic Admixture on Concrete Durability and Steel Corrosion"

_materials, 2022, doi:10.3390/ma15196842_

Round 1
Reviewer 1 Report
Authors are required to address these comments for the improvement of manuscript.
1. The lines from 24 to 26 in the introduction are confusing and long. Rewrite it to make it more meaningful.
2. Line no. 34 to 35 in Introduction how permeability will help to know about corrosion, justify? The introduction is weak. Authors must discuss corrosion problems and how past researchers (literature) have addressed this issue and include an introduction related to nano.In lines 37 to 39, in the introduction, how the author can claim that cement hydrophobic is the most effective method to improve the reinforced durability of concrete. What about other techniques? The author needs to include a paragraph about different techniques and should finally state the difference between this hydrophobic cement technique and other techniques. To strengthen the introduction part, author can include these papers. 1) doi.org/10.3390/ma15082728, 2) doi.org/10.3390/polym14153132, 3) doi.org/10.1016/j.compositesb.2022.110104
3. What is the full form of HRB in line no.69?
4. The average size of nano–hydrophobic is 60 nm. How did the author arrive at this value justify?
5. Section 2.3, refer to line no. 95, on what basis did the author select the size of the specimen? Section 2.3, line no. 100, In dry and wet cycle tests, why was the testing temperature selected at 50 °C, and how much depth were the specimens immersed in NaCl solution?
6. A picture of the specimens and their change in surface texture must be included when specimens are exposed to the solution.
7. In section 2.4.2, what was the temperature and humidity of the laboratory in which experiments were carried out?
8. In section 2.4.3, the codal provision is missing.
9. In sections 3.1, 3.2, and 3.3, the discussion part needs to be supported with literature.
10. Add a flow chart of the methodology.
11. From Lines 297 to 299, how did the authors conclude this statement?
12. What about slump tests? Why did the author missed it?
13. Why did the author not mention the physical properties of the ingredients, and especially steel?
14. In sections 3.1, 3.2, and 3.3, equipment names along with their specimens and testing conditions must be written.
15. A list of abbreviations must be included.
16. Over all authors have not discuss the analysis in details, why?
Author Response
Answer to Q1 and Q2: The introdution had been completely revised.
Answer to Q3: the HBR composition had been added in table 1.
Answer to Q4: The particle size through were test by laser particle size tester, which is indicator of products.
Answer to Q5: The specimens size reference to stardard of china, the temperature is keep the cement hydration products stable.
Answer to Q6: The picture had been modified.
Answer to Q7: The tmeperature and humidity of laboratory is 25 ℃ and 50% respectively.
Answer to Q8: the missing had been supplement.
Answer to Q9: The literature had been supported.
Answer to Q10~Q12: Some test had been conduted in ohter work.
Answer to Q13: the present study was mainly focused the steel corrosion influenced by TIA.
Answer to Q14-Q16: The some section had been modified.

Reviewer 2 Report
The authors present their work on the Influence of a nano-hydrophobic admixture on concrete durability. It is hard to follow the story because the sentence structure of this manuscript needs revision. I recommend the authors engage the service of a language editor to improve this paper.
Introduction
Line 24 -33: hard to understand. The durability of concrete... unstability of the passive film. Full of spelling and sentence structure poor.
Lines 34-42, similarly cannot be followed.
2 Materials and methods:
Figure 1 seems incomplete. It requires annotation and additional detail to show the electrical conductivity in the embedded steel.
2.4.3 Chloride difussion test. Line 130 -134
There is not enough details in this description. How was titration used to detect Cl concentration with depth? Are samples taken from different depths? How was the sample sectioned? Contamination?
2.4.4 Electrochemical measurement
Further details on the description of electrochemical tests. Including schematic illustrations will improve the readability.
Figure 5; x-axis should start from 0 since there is no negative depth.
Author Response
1. The introdution section have been completely revised.
2. materials and methods also compeletely modified.
Reviewer 3 Report
Authors of the publication "‘Influence of a nano-hydrophobic admixture on concrete durability and steel corrosion ” presented novel nano-hydrophobic admixture was introduced to improve the ion's diffusion properties and the corrosion resistance of reinforced steel in the reinforced steel concrete. The nano-hydrophobic admixture used in the study is an aqueous anionic microemulsion of hydrophobic nano-silica (TIA). Each of the presented parts of the publication has been relatively correctly described by the authors. The conclusions are consistent and closely related to the research topic. As a reviewer of this work, however, I believe that the reviewed work requires many corrections, which will undoubtedly improve its quality:
1. All sections should be revised. The method of reporting the procedure is not appropriate.
For example: "The weighing process should be completed within 15 s." Ended at this time?
Another example: "... rub with a dry towel to remove any residual water on the surface ...". Is the use of the towel important news?
2. There isn't a table containing the chemical composition of steel.
3. Table 2 should be on one page. Please remove the color.
4. Diagrams should be down the cited text.
5. Please don't use the shortcut of "day".
6. Figure 1 adds nothing. Perhaps it is worth adding a photo, e.g. with the dimensions of the sample and the location of the rebar steel and copper wire marked with it.
7. Figures 3 and 4 should be enlarged.
8. The water absorption research were carried out during 30 min, 300 min, 600 min, 1440 min, why at the same times no studies on water evaporation were carried out?

Author Response
A1: All section had been modified.
A2: table of steel compositon have been added.
A3~7: some section have been resvised.
A8: The water adsorption and water evaperatoin process are different, so the detection time was different.
Reviewer 4 Report
Influence of a nano-hydrophobic admixture on concrete durability and steel corrosion
Manuscript Number:
In the present paper authors provide an experimental investigation on the Influence of a nano-hydrophobic admixture on concrete durability and steel corrosion. However, the paper requires some major improvement before it can be recommended for publication, it is proposed to re-submit a thoroughly revised version of the manuscript, considering the following comments.
1. Title and abstract are ok 2. Overall recommendation should be reported in one sentence at the end of the abstract 3. The authors should overview the recent progress made in the relevant area in the past two years or so. 4. Emphasizing the importance of research in introduction 5. Please use space before units. Check whole manuscript 6. Please compare your results discussion with relevant studies 7. The paper is well written and it is easy to follow, only the authors needs to go thoroughly revised version to correct the typo-mistake.
8. Author should highlight the assumptions and limitations and future research direction of the study.
Author Response
A2: Overall recommendation have been added at the end of the abstract
A3-4: The introduction had been compeletely revised and include the recent progress made in the relevant area in the past two years .
A5: all of paper had been check, thanks.
A6: Discussion had been modified.
A7: ALl of paper had been revised.
A8: The assumptions and limitations and future research direction of the study had been added .
Round 2
Reviewer 1 Report
Manuscript ID: materials-1890792
Title: Influence of a nano-hydrophobic admixture on concrete durability and steel corrosion
Reviewer Comments: The above-revised manuscript addresses all my comments and can be accepted for publication.
Reviewer 2 Report
The authors have substantially improved on the earlier version by addressing recommended revisions.
Reviewer 3 Report
Authors of the publication "Influence of a nano-hydrophobic admixture on concrete dura- 2 bility and steel corrosion” presented the influence of the research on cement with the admixture of is an aqueous anionic microemulsion of hydrophobic nano-silica to improve the ions diffusion properties and the corrosion resistance of reinforced steel.
Each of the presented parts of the publication has been correctly described by the authors. The conclusions are consistent and closely related to the research topic. Each of the graphs is presented in a very legible and clear way, as well as their interpretation is described in detail. The research methodology presented is good. Summing up, the reviewed work presents a very high substantive and experimental value.
Reviewer 4 Report
The author addressed most of all reviewer comments